# Reply to "Basal buoyancy and fast-moving glaciers: in defense of analytic force balance" by C. J. van der Veen (2016)

Terence J. Hughes, Professor Emeritus of Earth Sciences and Climate Change
University of Maine

404 North Sixth Street, Fort Pierre, South Dakota 57532, U.S.A.

**Abstract.** Two approaches to ice-sheet modeling are available. Analytical modeling is the traditional approach. It solves the force (momentum), mass, and energy balances to obtain three-dimensional solutions over time, beginning with the Navier-Stokes equations for the force balance. Geometrical modeling employs simple geometry to solve the force and mass balance in one dimension along ice flow. It is useful primarily to provide the first-order physical basis of ice-sheet modeling for students with little background in mathematics (Hughes, 2012). The geometric approach uses changes in ice-bed coupling along flow to calculate changes in ice elevation and thickness, using floating fraction $\phi$ along a flowline or flowband, where $\phi = 0$ for sheet flow, $0 < \phi < 1$ for stream flow, and $\phi = 1$ for shelf flow. This leads to confusion in reconciling the two approaches (Van der Veen, 2016). An attempt is made at reconciliation.

## Introduction

Cornelis "Kees" Van der Veen's comparison of geometric and analytic approaches to the force balance in glaciology in *The Cryosphere* (Van der Veen, 2016) is most welcome because he takes seriously my geometrical approach to the longitudinal force balance, citing many of my paper from when I first introduced the concept (Hughes, 1992) to the latest application (Hughes et al., 2016). To begin, the analytic force balance is not challenged. The geometric force balance is useful only for one-dimensional flow along ice-sheet flowlines or flowbands of constant width. For two-dimensional flow in the map plane, width become a variable and geometrical areas become geometrical volumes; substantially increasing geometrical complexity with little advance in physical insight. The analytic force balance is typically obtained by solving the Navier-Stokes equations, which can be done in three dimensions and, when including the mass and energy balances, becomes time-dependent. The geometrical approach is useful for understanding the force balance by comparing the areas of right triangles and rectangles (or parallelograms).

## Problems with Van der Veen (2016)

My concerns are with his figures. His Figure 1 is fine, but his Figure 2 compares apples and oranges, a longitudinal stress gradient with basal and side drag stresses and a gravitational driving stress. A stress is not the same as a stress gradient but it allows Van der Veen (2016) to claim my gravitational "pulling stress" (Hughes, 1992) acts in the same direction as the gravitational driving stress. His stress gradient should be a force gradient, which then has units of stress. My pulling stress is an actual stress, the longitudinal tensile stress, not a longitudinal stress gradient. The pulling stress exists from the calving front to the grounding line of an ice shelf and up ice streams that supply the ice shelf. The pulling stress at the calving front of an ice shelf was derived analytically by Weertman (1957) and geometrically by Robin (1958).

Van der Veen's Equation (7), which he states is plotted in his Figure 2 for Byrd Glacier,
is correct but his plot is not. His "gradient in longitudinal stress" in his Figure 2 should be a
longitudinal gradient in the longitudinal force, $\partial(HR_{xx})/\partial x$, which is the sum of ice
thickness times the longitudinal gradient in longitudinal stress, $H\partial R_{xx}/\partial x$, plus the
longitudinal stress times the longitudinal gradient in ice thickness $R_{xx}\partial H/\partial x$, where $HR_{xx}$
is the longitudinal force, with longitudinal stress $R_{xx}$ averaged through $H$.

Van der Veen (2016) states his Equations (13) through (15) can be entered into his
Equation (9), and they are my equations for these stress terms and my balance equation,
but they are not. For my equations, see Table 12.1 in Hughes (2012), reproduced here as
Table 1. Substitute his Equations (13) through (15) into his Equation (9) and you will not
get 0 = 0, but you will if you use my equations in Table 12.1. His equations have both terms
and signs different from mine in my geometrical force balance. So his plots of his Equations
(13) through (15) in his Figure 2 are meaningless, both in terms of his own analysis and as
a critique of my geometrical force balance.

My longitudinal geometrical force balance at any distance *x* upstream from the ice-shelf
grounding line is shown geometrically in Van der Veen's Figure 3 when AF is parallel to BE
(my Figure 1). The gravitational driving force given by area ADF is balanced by resisting
forces given by areas ABEF, BCE, and CDE (areas 1, 2, 3, and 4 in my Figure 2), all of which
vary with floating fraction $\phi$ along *x* and sum to give area ADF. In my Figure 2 (bottom),
resistance forces over distance $\Delta x$ are given by the difference between areas 5 and 1 for
basal drag and areas 6 and 2 for side drag in the grounded fraction of ice, and the difference
between areas 7 and 3 for water buttressing and areas 8 and 4 for tensile pulling in the
floating fraction of ice. All resisting forces vary with $\phi$ along *x*. This is in agreement with
Van der Veen (2016) that resisting forces are calculated over $\Delta x$ in the longitudinal force
balance. It is impossible to get the geometrical force balance wrong if these simple rules are
followed.
My flotation, basal drag, and side drag stresses all act opposite to my driving stress, his do
not. Mine must, to complete the geometrical force balance. Readers of *The Cryosphere* can
see the geometric force balance applied to the calving front of an ice shelf and to a fully
grounded ice sheet on a flat bed, both derived geometrically, in Appendix A of Hughes et al.
(2016). These are the simplest applications that anyone who knows the area of a triangle is
half the height times the base can understand, the height being ice or water height and the
base being ice or water basal pressure. Van der Veen (2016) sees these applications for
sheet and shelf flow, but not for stream flow.

Van der Veen (2016) states my $F_g$ in his Equation (16) is not a longitudinal
gravitational driving force, but it is. Pressure has no direction so to get a longitudinal force
along ice flow it has be multiplied by the transverse cross-sectional area, which is variable
ice height for constant ice width. Hence, for basal ice pressure $P_I$ the gravitational driving
force is average ice pressure $\bar{P}_I$ times ice height $H$, which is the area of triangle ADF in his
Figure 3, which is reproduced as my Figure 1 (left) for comparison with my Figure 2, which
shows the correct geometry, Figure 5 in Hughes et al. (2016).
Figure 1 (left), Figure 3 in Van der Veen (2016), indicates he does not understand the
geometrical force balance for ice streams. Line AF should be parallel to line BE because
they both show how ice pressure increases with depth. Line CE shows how water pressure
increases with depth, as is obvious at the calving front. In the geometrical force balance, the
longitudinal gravitational driving force is area ADF of the big triangle. Fitted inside ADF are
a resisting flotation force given by area BDE for the floating ice fraction and a resisting drag
force given by area ABEF for the grounded ice fraction. Inside BDE is area CDE for the
resisting force from water pressure and area BCE for the resisting force from the tensile
strength of ice. Inside ABEF is the triangle above B for basal drag and the parallelogram
below B for side drag. Resistance from basal drag is the area of the triangle above B.
Resistance from side drag is the area of the parallelogram below B if lines BE and AF are
made parallel. If BE is made part of AF a rectangle would replace the parallelogram but the
area would be unchanged, see my Figure 2. That's all there is to it. The only remaining task
is to replace forces with products of stresses and lengths (areas having unit or fixed widths
along $x$) upon which the stresses act along a flowline (no width) or a flowband (constant
width). My solution for the force balance is exact. All gravitational and resisting forces in
the longitudinal direction of ice flow are included.
For example, at distance $x$ from the ice-shelf grounding line in Figure 2, gravitational
driving force $F_G = \bar{P}_I h_I$ is resisted by the sum of the upstream tensile pulling force
$F_T = \sigma_T h_I$ and the downstream compressive pushing force $F_C = \sigma_C h_I$ so $\sigma_T = \bar{P}_I - \sigma_C$. Here
resisting force $\sigma_C h_I$ is balanced by the gravitational force given by areas 1+2+3 in Figure 2
(center and bottom), and includes all downstream resistance due to averaged basal and
side shear stresses $\bar{\tau}_O$ and $\bar{\tau}_S$ respectively linked to gravitational areas 1 and 2, plus local
water stress $\sigma_W$ linked to area 3.
The major variable in the geometrical force balance is the floating fraction $\phi$ of ice,
where $\phi = 0$ for sheet flow, $0 < \phi < 1$ for stream flow, and $\phi = 1$ for shelf flow. Here we are
primarily interested in stream flow as shown in my Figure 3. From Newton's second law of
motion in a vertical force balance, gravitational force $F_G$ at the base must be the same for
floating area $w_F \Delta x$ and total area $w_I \Delta x$ such that $F_G = (\rho_I h_I w_F \Delta x)g = (\rho_I h_F w_I \Delta x)g$ for ice
density $\rho_I$ and gravity acceleration g to obtain basal pressures $P_F = \rho_I g h_F$ and $P_I = \rho_I g h_I$
that support ice of respective floating and total heights $h_F$ and $h_I$. This vertical force
balance is satisfied if $h_F$ goes from 0 to $h_I$ as $w_F$ goes from 0 to $w_I$. The basal water
pressure is $P_W = \rho_W g h_W = P_F = \rho_I g h_F$ for water density $\rho_W$ and water height $h_W$ needed to
float ice height $h_F$. The floating fraction of ice at $x$ is therefore:
$$\phi = w_F / w_I = h_F / h_I = P_F / P_I = P_W / P_I.$$
Pulling force $\sigma_T h_I$ resists the gravitational driving force given by area 4 in Figure 2
(bottom), which is area 3+4 minus area 3. Area 3+4 is one-half flotation height $h_F = h_I \phi$
times basal floating length $P_F = P_I \phi$, so area 3+4 is $\bar{P}_I h_I \phi^2$. Area 3 is one-half height
$h_W = (\rho_I / \rho_W) h_F = (\rho_I / \rho_W) h_I \phi$ times the same basal floating length $P_F = P_I \phi$. Then the
tensile pulling stress is $\sigma_T = \bar{P}(1 - \rho_I / \rho_W)\phi^2$. It is that simple. At the calving front where
$\phi = 1$ this is the solution obtained by Weertman (1957) and Robin (1958). Table 1 lists all
stresses resisting gravitational forcing at *x*.

Figure 1(right) shows Figure 4 in Van der Veen (2016). His Figure 4(a) is too simplistic.

If it were true there would be no thinning of a flat ice shelf or at ice divides of an ice sheet
because neither has a surface slope. Yet thinning of both occurs. For ice shelves the correct
analytical solution was provided by Weertman (1957, Appendix). Hughes (2012a, Chapter
9) provided the correct geometrical solution even if the ice shelf has a thickness gradient in
the flow direction. Raymond (1983) provided the correct analytical solution for ice divides.
The gravitational driving stress in Van der Veen's Figure 4(a) is zero because his
longitudinal arrows that lengthen with depth *z* cancel each other from top to bottom.
Instead, his Figure 4(a) shows the vertical force balance in which the downward
gravitational force in the *z* direction is the mass of overlying ice times the vertical
acceleration of gravity, and it is balanced by the upward pressure of ice acting on unit area
in the horizontal *xy* plane at any depth *z* below the ice surface from top to bottom.

Van der Veen's Figure 4(a) cannot represent the tensile longitudinal deviator stress, my

pulling stress, for both ice shelves and ice divides. The two triangles have equal areas so
there can be no longitudinal spreading in his way of thinking because there is no ice surface
slope. For an ice shelf, one of his triangles should be moved to the calving front. Then he
would see the pulling force in action because a water triangle would replace his ice triangle.
For an ice divide, downslope motion on opposite flanks of the ice divide produce a
longitudinal tensile stress under the ice divide, and that ice thinning lowers the ice divide.

Figure 1(right) also shows Figure 4(b) in Van der Veen (2016), which has a surface

slope, causing a difference in area of his two ice triangles. This difference is his
gravitational driving force for sheet flow, which is balanced by basal drag that requires a
basal shear stress applied along length $\Delta x$ between the triangles as a drag force. There is
no basal drag under an ice shelf, except where surface ice rumples appear above basal
pinning points, see my Figure 2. For stream flow, Figure 2 gives the correct geometrical
representation of gravitational forcing in the longitudinal direction *x* of ice flow.

Van der Veen (2016) repeatedly refers to my 2008 unpublished research report, which

is not readily available. More complete and better treatments are in Hughes (2012a) and
Hughes et al. (2016). Van der Veen states, "Balance of forces is only meaningful if applied to
flow-line segments, not single locations. Consequently, the concept of force balance at any
location is inherently flawed." Not true. The balance is meaningful at the calving front of an
ice shelf, a single location (Hughes et al., 2016, Appendix A) and at any upstream point by
including a local compressive stress $\sigma_C$ which includes downstream resistance to ice flow
all the way to the calving front, see Figure 2 (middle), and Equations (11) and (19) in
Hughes et al. (2016).

I agree with Van der Veen (2016) that longitudinal stress gradients are important, and I

include downstream resistance to ice flow in my force balance at any point location, see
Figure 2 (top). Resisting stresses at that point are in Table 12.1 of Hughes (2012a) and are
Equations (11) through (18) in Hughes et al. (2016). My longitudinal stress gradients
include basal and side shear stresses averaged over the downstream length to the calving
front of a linear flowband, see Table 12.1, divided by the corresponding downstream
flowband length, for sheet ($\phi = 0$), stream ($0 < \phi < 1$), and shelf ($\phi = 1$) flow, where $\phi$ is
the floating fraction of ice in Van der Veen (2016), and is my $\phi$.
Referring to Hughes (2008), Van der Veen (2016) is incorrect in stating I believe lateral
drag vanishes at the center of a glacier. Figure 1 (left) is his Figure 3, and represents his
longitudinal gravitational driving forces along flow if his lines AF and BE are parallel. Then
his area ABEF is gravitational forcing resisted by both basal and side drag in an ice stream,
neither of which vanishes until the ice stream becomes a freely floating ice shelf without
basal and side drag, see Figure 6 in Hughes et al. (2016). Only when the solution is for a
flowline, not a flowband, does the side shear stress, representing lateral drag, vanish. My
correct counterpart to Figure 3 in Van der Veen (2016) is Figure 2.
**The Geometrical Force Balance**
I developed the geometrical force balance to teach the fundamentals of glaciology to
students with an inadequate background in mathematics, usually students studying to be
glacial geologists, so my geometrical approach was designed to make maximum use of
glacial geology in reconstructing former ice sheets (Hughes, 1998, Chapters 9 and 10) and
in demonstrating how basal thermal conditions produce glacial geology under present-day
ice sheets (Hughes, 1998, Chapter 3). Previously I had spent more time teaching calculus
than glaciology because the Navier-Stokes equations had to be integrated in the force
balance.
My geometrical force balance is shown in Figure 2, which is Figure 5 in Hughes et al.
(2016). Along incremental length $\Delta x$, change $\Delta F_G$ in the longitudinal gravitational driving
force $F_G$ is balanced by change $\Delta F_T$ in the tensile pulling force $F_T$ plus change $\Delta F_W$ in the
water buttressing force $F_W$ plus basal drag force $F_O$ plus side drag force $F_S$, where
$F_F = F_T + F_W$ is a flotation force that requires ice-bed uncoupling by basal water. Dividing
by $\Delta x$ and letting $\Delta x \rightarrow 0$ gives as the longitudinal gravitational force gradient
$$\partial F_G / \partial x = \partial(\bar{P_I} h_I) / \partial x = P_I \alpha_I = \partial(\sigma_F h_I) / \partial x + \tau_O + 2\tau_S(h_I / w_I)$$
where the bed is represented by an up-down staircase with successive $\Delta x$ steps so ice
thickness gradient $\alpha_I$ equals $\alpha$ for ice surface slope on each step, $P_I$ is the overburden ice
pressure at the base, $\tau_O$ is the basal shear stress, $\tau_S$ is the side shear stress for two sides,
$h_I$ is ice thickness, $h_W$ is the height of water that floats flotation height $h_F$ of ice supported
by basal water pressure $P_W$ such that $P_W = P_F$ and $h_W = (\rho_I / \rho_W)h_F$ for floating fraction $\phi$,
and $\sigma_F = \sigma_T + \sigma_W = \bar{P_I}\phi^2$ for ice tensile stress $\sigma_T$ and water buttressing stress $\sigma_W$, all at
distance $x$ upstream from an ice-shelf grounding line. At the calving front of an ice shelf
where $\phi = 1$ so $h_F = h_I$ this is identical to the Weertman (1957) and Robin (1958)
solutions. Together $\sigma_T$ and $\sigma_F$ resist gravitational forcing $\bar{P_I}$ in an ice shelf and $\bar{P_I}\phi^2$ due to
floating fraction $\phi$ in an ice stream at $x$. My $\sigma_F$ would be $R_{xx}$ in Equation (1) of Van der
Veen (2016), taking account of the different sign conventions, except my $\sigma_F$ always
requires basal water that uncouples ice from the bed. In ice streams, water height $h_W$
above the bed is the height to which water would rise in a borehole (Kamb, 2001).
Resistance from my $\sigma_W$ may be akin to bridging stresses across water-filled cavities
discussed by Van der Veen (2016). The existence of $\sigma_W$ in the geometric force balance is
not readily apparent from analytic solutions of the Navier-Stokes equations, but Van der
Veen (2016) may have teased it out with his bridging stress, which forces him to add
resistance by including steep shear-stress gradients on each side of his cavities. He
maintains his cavities are small so these gradients average out to zero along an ice stream,
eliminating the need for my $\sigma_W$. They cannot average to zero if his cavities are water-filled
and get bigger and closer together downstream, as required to progressively uncouple ice
from the bed. Then cavities themselves have a size and distribution gradient. Figure 3,
which is Figure 4 in Hughes et al. (2016), shows my concept of water-filled cavities in area
$w_I \Delta x$ under an ice stream. The plain fact is we do not know which concept of cavities is
correct.
I developed the geometrical force balance over some decades, from Hughes (1992)
through Hughes et al. (2016). My papers are a work in progress, see pages 201-202 of
Hughes et al. (2016) regarding $h_W$, $h_F$, $\sigma_W$, and $\sigma_F$ not included in earlier papers. To
access my most recent thinking, see Hughes (2012) and Hughes et al. (2016). All the earlier
studies are flawed in various ways. The last ones may also have flaws I haven't detected.
Criticisms by Van der Veen (2016) are mainly directed at my earlier flawed papers.
This response gives me an opportunity to correct three mistakes in Hughes (2012a).
They will be obvious to the careful reader. The first line in Equation (12.9) should be:
$$\partial(\sigma_F h_I)/\partial x = \partial\left[\frac{1}{2}\rho_I g h_I^2 \phi^2\right]/\partial x = P_I \phi(\phi\alpha_I + h_I \partial\phi/\partial x)$$

and in the second line $\phi$ should be $\phi^2$. In the denominator of Equation (17.18), $r$ should be
replaced by $(a - r)$. The first line of Equation (22.18) should be:
$$\Delta h_i^* / \Delta x = \phi^2\left(\frac{\Delta h_I}{\Delta x}\right)_i + \left(\frac{h_I}{2}\right)_i \frac{\Delta\phi^2}{\Delta x} + \frac{(\tau_O)_i}{\rho_I g h_I^*} + \frac{2(\tau_S)_i}{\rho_I g w_I} = \frac{(\tau_O^*)_i}{\rho_I g h_I^*}$$

Equation (22.18) applies to sheet flow, for which $\phi = \partial\phi/\partial x = 0$ and $\tau_O^*$ increases
resistance from basal drag $\tau_O$ by including side drag $\tau_S$ in flowbands having some side
shear. Since tributaries supplying ice streams are ubiquitous in the sheet-flow interior of
the Antarctic Ice Sheet (Hughes, 2012b), and tributaries are flowbands, side shear must be
taken into account even for sheet flow.
**Concluding remarks**
May I conclude with some general observations? Suppose an iceberg were released
where the two equal triangles meet in Figure 1 (right). This is Figure 4(a) in Van der Veen
(2016) for his ice shelf. He would have us believe the force balance was suddenly
transformed to the balance analyzed by Robin (1978) at the calving front for the same ice
thickness. But the force balance does not change. Gordon Robin also did not understand
this. I submitted my manuscript, "On the pulling power of ice streams" to the Journal of
Glaciology in 1988. Gordon rejected it on the grounds that the geometrical force balance he
used at the calving front didn't apply back to the grounding line and up ice streams that
supply the ice shelf because water height $h_W$ existed only at the calving front. My reply to
that is given on pages 201-202 of Hughes et al. (2016). I had given my 1988 manuscript to
Mikhail Grosswald and he showed it to Russian glaciologists, resulting in an invitation to
present my geometrical force balance to the U.S.S.R. Academy of Sciences. In case Gordon
had spotted a fatal flaw, on my way to Moscow I stopped in Cambridge to discuss it with
Gordon and Charles Swithinbank. Charles understood the concept. Gordon did not; he just
"knew" the concept had to be wrong. My manuscript was finally published four years later
through the efforts of Garry Clarke as Editor-in-Chief (Hughes, 1992).
I had the same experience with Johannes Weertman. When I presented my "theory of
thermal convection in polar ice sheets" at a 1975 symposium of the International
Glaciological Society (Hughes, 1976), Hans told me, "I feel in my bones it doesn't happen." I
replied, "Let me know when you hear from your brain." Well, it still hasn't "happened" even
when it seemed to me the evidence was staring us right in the face (Hughes, 1985).
Weertman's "bones" may be more reliable than Hughes' brain. Be that as it may, now I
believe thermal convection rolls underlie tributaries of ice streams, which are ubiquitous
on the Antarctic Ice Sheet, and I have recommended field tests of this idea (Hughes, 2012b).
Here's another example from my half-century in science: The International Glaciological
Society reviewers didn't like the way I used glacial geology to reconstruct ice sheets at the
Last Glacial Maximum 18,000 years ago from the bottom up for CLIMAP (Climate: Long-
range Investigation, Mapping, and Prediction) in 1980, so George Denton and I published
our CLIMAP work as a book (Denton and Hughes, 1981). The book is now a classic. The
bottom-up geometrical approach using glacial geology can also be used to reconstruct ice
sheets for a whole glaciation cycle (Hughes, 1998, Chapters 9 and 10), for comparison with
ice sheets reconstructed using the analytical approach for a glaciation cycle (Fastook and
Hughes, 2013), and for deducing glacial geology produced under the Antarctic Ice Sheet
today by mapping basal thermal zones from ice thicknesses and elevations along surface
flowlines (Hughes, 1998, Chapter 3; Wilch and Hughes, 2000; Siegert, 2001).
Cornelis van der Veen understands ice dynamics as well as anyone, so I am left with the
puzzlement expressed by the Apostle Paul in Acts 28:26. "You may listen carefully yet you
will never understand; you may look intently yet you will never see." He is not alone.
Reviewers of his paper also did not see the obvious. Maybe it is obvious only to me.
*Acknowledgements.* I thank Cornelis van der Veen for giving me the opportunity to further
explain the geometric force balance in relation to the analytic force balance.

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

**Table 1**: *Resisting Stresses Linked to Floating Fraction* $\phi = P_F/P_I$ *of Ice and Gravitational*
*Forces Numbered in Figure 2 for the Geometrical Force Balance.*

| |
|---|
| Basal water pressure at *x*, from gravity force 3:<br>$$P_W = \rho_W g h_W$$ |
| Ice overburden pressure at *x*, from gravity force (1+2+3+4):<br>$$P_I = \rho_I g h_I$$ |
| Upslope tensile stress at *x*, from gravity force 4:<br>$$\sigma_T = \bar{P}_I \left(1 - \rho_I / \rho_W\right)\phi^2$$ |
| Downslope compressive stress at *x* due to $\bar{\tau}_O$ and $\bar{\tau}_S$ along *x* and $\sigma_W$ at *x* = 0:<br>$$\sigma_C = \bar{P}_I - \sigma_T = \bar{P}_I - \bar{P}_I\left(1 - \rho_I / \rho_W\right)\phi^2$$ |
| Downslope water-pressure stress at *x*, from gravity force 3:<br>$$\sigma_W = \bar{P}_I \left(\rho_I / \rho_W\right)\phi^2$$ |
| Upslope flotation stress at *x* from gravity force (3+4):<br>$$\sigma_F = \sigma_T + \sigma_W = \bar{P}_I \phi^2$$ |
| Longitudinal force balance at *x* from gravity force [(5+6+7+8)–(1+2+3+4)]:<br>$$P_I \alpha = \partial\left(\sigma_F h_I\right)/\partial x + \tau_O + 2\tau_S \left(h_I / w_I\right)$$ |
| Flotation force gradient at *x* from gravity force [(7+8)–(3+4)]:<br>$$\partial\left(\sigma_F h_I\right)/\partial x = P_I \phi\left(\phi\alpha_I + h_I \partial\phi / \partial x\right)$$ |
| Basal shear stress at *x* from gravity force (5–1):<br>$$\tau_O = P_I \left(1 - \phi\right)^2 \alpha - P_I h_I \left(1 - \phi\right)\partial\phi / \partial x$$ |
| Side shear stress at *x* from gravity force (6–2):<br>$$\tau_S = P_I \left(w_I / h_I\right)\phi\left(1 - \phi\right)\alpha + \bar{P}_I w_I \left(1 - 2\phi\right)\partial\phi / \partial x$$ |
| Average downslope basal shear stress to *x* from gravity force 1:<br>$$\bar{\tau}_O = \bar{P}_I w_I h_I \left(1 - \phi\right)^2 / \left(w_I x + A_R\right)$$ |
| Average downslope side shear stress to *x* from gravity force 2:<br>$$\bar{\tau}_S = P_I w_I h_I \phi\left(1 - \phi\right)/\left(2\bar{h}_I x + 2 L_S \bar{h}_S + C_R \bar{h}_R\right)$$ |


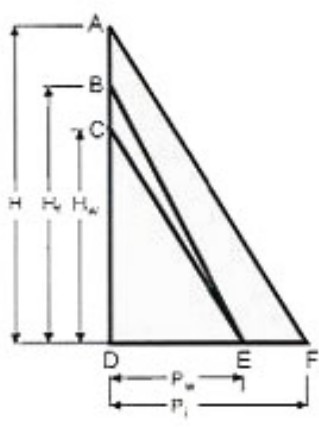 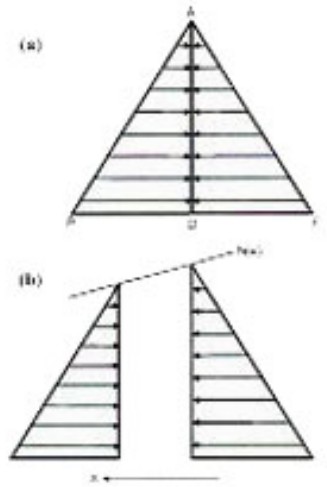


Figure 1: Figure 3 (left) and Figure 4 (right) from Van der Veen (2016).

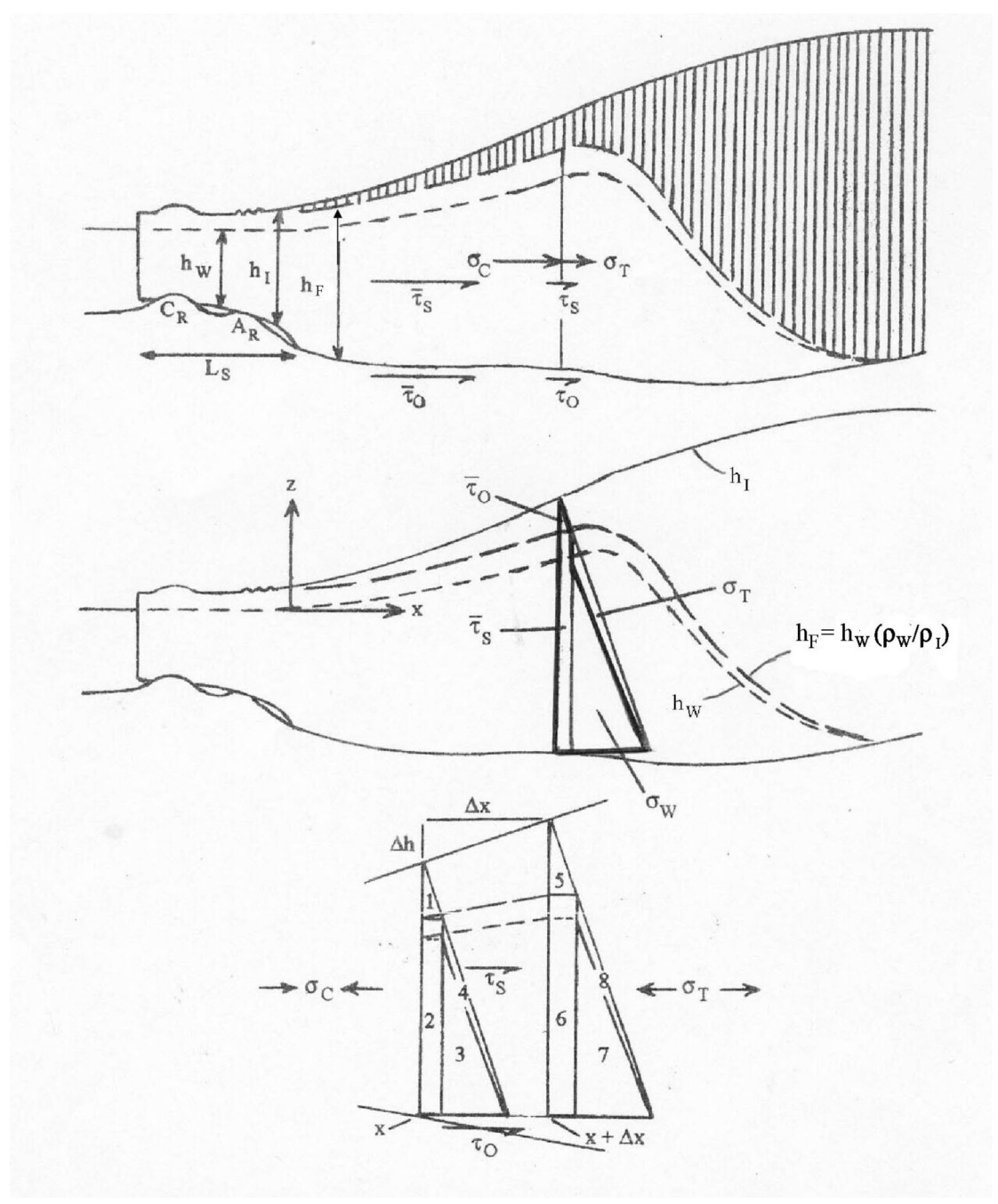


Figure 2: Figure 5 from Hughes et al.(2016). Top: Stresses at $x$ and downstream from $x$ that
resist gravitational forcing. The bed supports ice in the shaded area. Middle: The
gravitational force inside the thick border is linked to $\sigma_C$ which represents all downstream
resistance to ice flow at point $x$. Bottom: Gravitational forces (geometrical areas 1 through
8) and resisting stresses along incremental downstream length $\Delta x$ at point $x$.

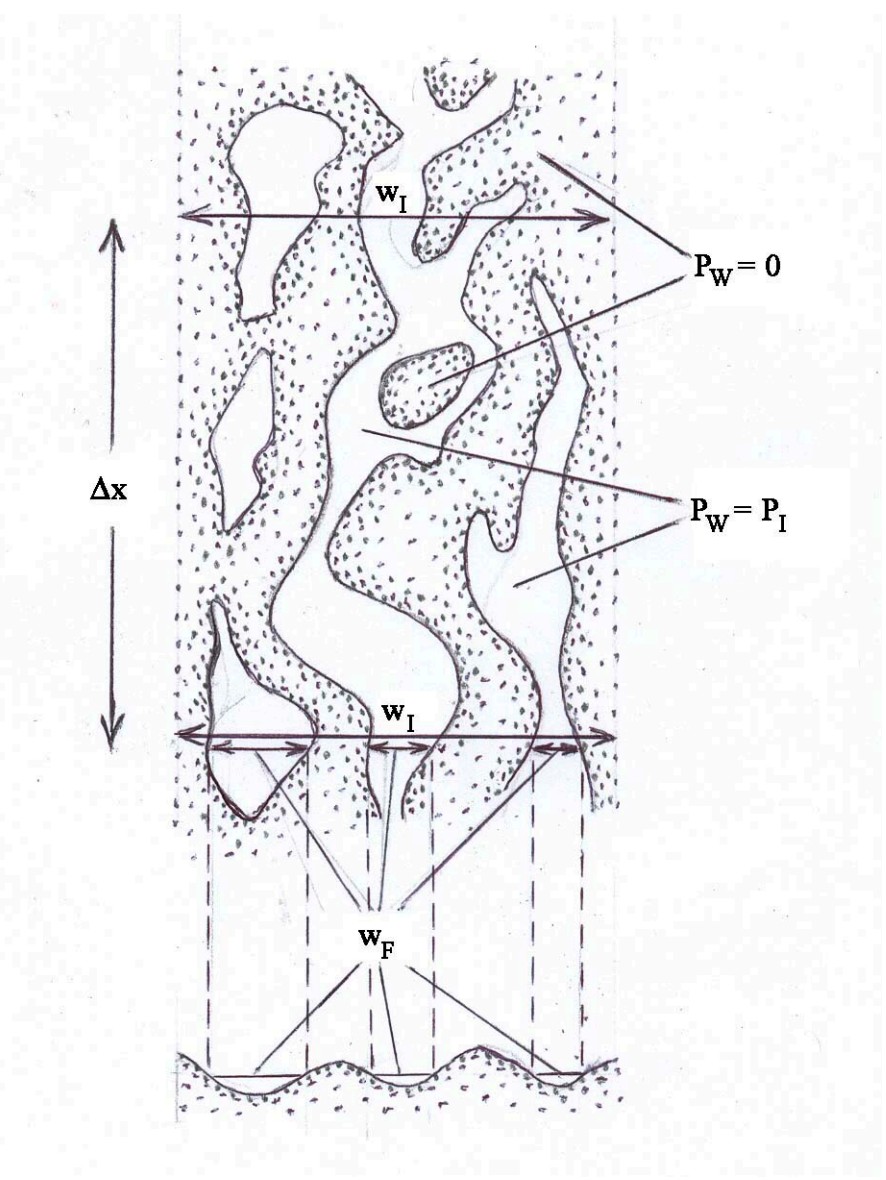


Figure 3: Figure 4 from Hughes et al. (2016). Under an ice stream, basal ice is grounded in
the shaded areas and floating in the unshaded areas (top) as seen in a transverse cross-
section (bottom) for incremental basal area $w_I \Delta x$.