# Peer review of "Reply to "Basal buoyancy and fast-moving glaciers: in defense of analytic force balance" by C. J. van der Veen (2016)"

_The Cryosphere, 2017_

## Referee Comment (RC1) · C. Van der Veen (Referee) · 27 Feb 2017

**Response to Prof T.J. Hughes reply in The Cryosphere Discussions, 31 January 2017.**

It is a tad disappointing that Prof. Hughes' reply to my note in The Cryosphere [Van der Veen, 2016] is so superficial and riddled with mistakes and apparent misconceptions about my more conventional approach. This suggests that the geometric force balance advocated by Prof. Hughes is, indeed, indefensible, as argued in my note. I will limit this response to the Section entitled "Problems with Van der Veen (2016)" – the following Section is a mere cut-and-paste from earlier work, and the "Concluding remarks" read like a lament on the half century of misunderstanding that Prof. Hughes apparently has suffered from reviewers and the scientific community. This may be of some interest from a historical perspective, but is irrelevant for the present discussion.

Prof. Hughes states that my equations (1), (2), (7) and (15) "can be confusing because they employ open parenthesis ) instead of closed parentheses () so terms are not properly separated." I have read this rather perplexing statement multiple times and still fail to understand its meaning or implication. Perhaps Prof. Hughes should check his pdf-reader to ensure it reproduces equations correctly on the screen or in print.

Equation (1) in my note is simply the definition of stress deviators; Equation (2) is the essential equation of my rebuttal and expresses force balance along a flowband or flowline. For sake of further discussion, this equation reads

$$
\tau_{dx} \;=\; \tau_{bx} \;-\; \frac{\partial}{\partial x}\!\left(H\tilde{R}_{xx}\right) \;-\; \frac{\partial}{\partial y}\!\left(H\tilde{R}_{xy}\right)\;. \tag{1}
$$

For Prof. Hughes benefit let me reiterate the meaning of the various terms in this equation. The left-hand side represents the gravitational driving stress, responsible for making the glacier flow in the downslope direction. The three terms on the right-hand side represent drag at the glacier base, $\tau_{bx}$,

resistance associated with gradients in longitudinal stress, also called longitudinal stress gradients,

$\frac{\partial}{\partial x}\left(H\tilde{R}_{xx}\right)$, and resistance associated with lateral drag, or friction originating at fjord walls or slowermoving ice, $\frac{\partial}{\partial y}\left(H\tilde{R}_{xy}\right)$. The tilde (~) denotes depth-averaged values of the resistive stresses, $R_{xx}$ and

$R_{xy}$. I am puzzled as to why this equation, which has been derived and applied many times before, can

be confusing. But Prof. Hughes following comment makes it clear that he fails to understand this simple

force-balance equation.

According to Prof. Hughes, my "Figure 2 compares apples and oranges, a longitudinal stress

gradient with basal and side drag stresses and a gravitational stress. A stress is not the same as a stress

gradient." Really? My Figure 2 shows the four force-balance terms as expressed by the above equation.

Both the longitudinal stress gradient term, and the lateral drag term, are estimated from measured

map-view *gradients* in the corresponding resistive stresses, $R_{xx}$ and $R_{xy}$. So, contrary to what Prof.

Hughes claims, my Figure 2 is correct and compared apples to apples (or oranges to oranges).

It might be instructive here to quote Turcotte and Schubert on the difference between Body Forces

and Surface Forces [Turcotte and Schubert, 2002, p. 73].

"Body forces acts throughout the volume of the solid. The magnitude of the body force on an

element is thus directly proportional to its volume or mass. An example is the downward force of

gravity, that is, the weight of an element, which is the product of its mass and the acceleration of gravity

g. [….] Surface forces act on the surface area bounding an element of volume. They arise from

interatomic forces exerted by material on one side of the surface onto material on the opposite side.

The magnitude of the surface force is directly proportional to the area of the surface on which it acts. It

also depends on the orientation of the surface."

In Section 2-3, Turcotte and Schubert [2002] proceed to consider the surface forces acting on a small rectangular element with dimensions δx, δy, and δz. This, of course, is the same derivation as I have given previously [Van der Veen, 2013, Section 3.1], and leads to the general balance of forces in the x-direction:

$$\frac{\partial \sigma_{xx}}{\partial x} + \frac{\partial \sigma_{yx}}{\partial y} + \frac{\partial \sigma_{yz}}{\partial z} = 0 \quad . \qquad\qquad (2)$$

Similar equations hold for the other directions, along the y- and z-axes. This fundamental form of the equation describing balance of forces can be used to derive the force-balance equation (1). Most importantly for the present discussion, it demonstrates that the concept of balance of forces as applied to glaciers has only physical meaning when applied to finite volumes of ice. Flow of glaciers is driven by *gradients* in the gravitational lithostatic stress that can only be estimated over a certain horizontal distance. The terms opposing this stress (the terms on the right-hand side of equation (1)) correspondingly apply to the same horizontal distance. Also, note how in equation (2) the balance of forces is expressed in terms of stress gradients.

The considerations given above do not imply that at any location in the ice sheet balance of forces is not satisfied. Of course it is, but this balance becomes rather meaningless with the lithostatic stress at any location balanced by an equal but opposite lithostatic stress, as shown in Figure 4a in Van der Veen [2016] – no matter how many times Prof. Hughes repeats assertions to the contrary. At the calving front of a glacier, where the (depth-averaged) lithostatic stress in the ice has to be balanced by the (depth-averaged) pressure exerted by the sea water, the familiar boundary condition used in numerical models is found [Van der Veen, 2013, eqs. (9.69) and (9.70)].

It is always a good idea not to put words in someone else's mouth. Case in point is the statement by Prof. Hughes, where referring to my Figure 4a, "if it were true there would be no thinning of a flat ice

shelf or at ice divides of an ice sheet because neither has a surface slope. Yet thinning of both occurs. For ice shelves the correct analytical solution was provided by Weertman (1957)." The reference to Weertman's solution for an ice shelf is somewhat surprising in this context because Weertman's Figure 1 clearly shows an ice shelf with zero surface slope, and Weertman explicitly states that "we now make the assumption that at a position far from the edge of the ice shelf all the stress components must be independent of the" two horizontal directions. As I have shown in the section on ice-shelf spreading [Van der Veen, 2013, sect. 4.5] this assumption of no surface slope or stresses independent of the flow direction, need not be made to arrive at the same solution as Weertman did.

More importantly is the contention by Prof. Hughes that no thinning will occur if the driving stress is zero, as on flat ice shelves or at ice divides. This need not be the case and I have never suggested such. An ice divide, or better, a flow divide, represents the vertical plane through which no transfer of mass takes place. Indeed, the local driving stress is zero and the horizontal velocity at that location is zero at all depths. There is still a vertical component of velocity and a vertical strain rate, however. Incompressibility requires the vertical strain rate to be balanced by a horizontal strain rate, or a horizontal velocity gradient. Consequently, there is local flow divergence, and for this condition to be in steady state, the surface mass balance has to be such that it compensates for this flow divergence. A similar line of argument can be applied to a flat ice shelf, in which the stretching rate, or velocity gradient, is constant in the flow direction.

As I stated at the beginning of this brief response, Prof. Hughes has failed to provide convincing or, for that matter, valid arguments to demonstrate that my derivation is wrong or seriously flawed. I have to conclude, therefore, that the concept of "pulling stress" as introduced and advocated by Prof. Hughes is nothing more than a red herring that, apparently, has obfuscated the thinking of Prof. Hughes. One must conclude that Weertman's "bones" proved correct after all.

C.J van der Veen

University of Kansas

February 27, 2017.

Turcotte, D.L., and G. Schubert (2002), *Geodynamics (Second Edition).* Cambridge UK, Cambridge University Press, 456 pp.

Van der Veen, C.J. (2013), *Fundamentals of Glacier Dynamics. (Second Edition).* Boca Raton FL: CRC Press (Taylor and Francis), 389 pp.

Van der Veen, C.J. (2016), Basal buoyancy and fast-moving glaciers: in defense of analytic force balance. *The Cryosphere 10*, 1331-1337.

Weertman, J. (1957), Deformation of floating ice shelves. *Journal of Glaciology 3*, 38-42.

---

## Author Comment (AC1) · 28 Feb 2017

Cornelis van der Veen has responded to my reply to his note in The Cryosphere [Van der Veen, 2016]. I am pleased to address his concerns. First, a minor point: The version of his paper I saw had only open parentheses ) associated with his equations, not closed parentheses(). But as I stated, that did not prevent me–or anyone else familiar with these equations–from closing the parentheses with ( in the conventional way. The flawed version was sent to me by my colleague, Professor James Fastook at the University of Maine. I don't know where he got it. I now see the version published in The Cryosphere is free from this defect.

Van der Veen's Figure 2 still confuses me. It compares a "driving stress" with a "gradient in longitudinal stress", "basal drag", and "lateral drag" but no dimensions are provided for numbers on the vertical axis. From the numbers, I assume the units are stresses, kilo-Pascals to be specific. But a "gradient" in kilo-Pascals needs units of length along flow. To me that's comparing apples and oranges. So his Figure 2 demonstrates nothing.

Van der Veen's Figure 3 does not represent my geometrical force balance because his lines AF and BE are not parallel. I can only assume he doesn't understand the geometrical force balance.

Van der Veen's Figure 4(a) is a vertical force balance, even though his horizontal arrows create the impression he wants it to be a horizontal force balance. However, forgetting the arrows, Figure 4(a) will not allow horizontal flow either on a flat ice shelf with no ice thickness gradient or down the flanks of an ice divide where the surface slope is zero. His Figure 4(b) with arrows and with a surface slope is correct.

Van der Veen does not seem to recognize that my floating fraction "phi" of ice along an ice stream is based on the vertical force balance, a body force, as quantified by Newton's second law of motion: force (of gravity) is mass (of ice) times acceleration (of gravity). I will make this unmistakably clear when I revise my version now available for interactive discussion.

Van der Veen does not have to "defend" the analytic approach to the force balance. I freely admit my geometric approach will never replace integrating the Navier-Stokes equations in the analytic force balance for serious modelers of ice-sheet dynamics. My approach is for students, primarily glacial geologists, who understand simple geometry but lack an adequate understanding of calculus.

I do maintain that the geometric force balance allows a deeper insight into the role of basal water under an ice stream that is deep enough to remove contact with the bed for some significant floating fraction of ice that generally increases downstream, and becomes complete ice-bed uncoupling for a freely-floating ice shelf.

Contrary to what Van der Veen states,I have never stated nor would I state that Van der Veen's treatment of the force balance "is wrong or seriously flawed" (his words). On the contrary, I stated "Cornelis van der Veen understands ice dynamics as well as anyone..." My point then and now is none of us has a complete understanding of the force balance in glaciology, or of any other aspect of glaciology for that matter. Glaciology is not "settled science" unlike climate science–as we are repeatedly told.

More than most, my career in glaciology has been marked by challenging conventional thinking. I mentioned three other examples in addition to the geometrical force balance: (1) the "pulling power" of ice streams, (2) the possibility of thermal convection in polar ice sheets, (3) former ice sheets can be reconstructed from the bottom up using glacial geology more reliably than from the top down using the mass balance. There are many others. All have been published in refereed journals, with the assistance of reviewers who are worth their weight in gold.

Terence J. Hughes Fort Pierre, South Dakota 28 February 2017

---

## Referee Comment (RC2) · Anonymous Referee #2 · 17 Mar 2017

This has been a troublesome review to prepare - refereeing a dispute between a pair of senior scientists.

Hughes has developed what he calls the geometrical force balance, by his account a novel and simplified way of understanding the controls on glacial flow. van der Veen carefully pointed out the flaws in this approach, which neglects fundamental aspects of the balance of forces at play in glacial ice. Both the geometric force balance and van der Veen's criticisms of it have been published for reference. Now, I gather Hughes would like to criticize the criticisms of the validity.

van der Veen reasons about the mechanics of flow using well established, mathematical constructs; e.g. from Stoke's flow, it follows that...Having read Hughes' criticisms, I

still can not find fault with van der Veen's approach, his reasoning, or his conclusions. Hughes, on the other hand, appears to be reasoning more intuitively system while offering a defense of it. In many cases he reveals his own misconceptions, or misplaced antagonism about superficial issues, like van der Veen's equation formatting.

In a way, maybe the review should come down to this: I have not been persuaded by Hughes' rebuttal. I find van der Veen's criticisms and rebuttal to the rebuttal very persuasive.

However, Hughes has achieved too much and demonstrated unique insight too often to be ignored. His intuitive approach animates the community and helps set the scientific agenda - often through provocative catch phrases or articulation of very high level processes. van der Veen, has also been very influential in glaciology. His approach is methodical and builds upon centuries of mathematical analysis. He writes the textbooks we all study in order to better understand the controls on glacial flow. Much of his work carefully isolates components of the mechanical controls on flow and makes quantitative comparisons of their importance.

Given that sketch of the two parties involved, it's not surprising they are at odds over Hughes' geometrical force balance. But, given that it is published, the subsequent debates over its validity provide and interesting context for how ideas are generated and debated in science. I'm favor of 'publishing' (electronically) the entire exchange in hopes that subsequent generations of scientists can see what a messy affair this whole business can be.

---

## Author Comment (AC2) · 18 Mar 2017

The exchange between C.J. van der Veen and me boils down to his belief there is only one way to skin a cat for the force balance in glaciology, integrating the Navier-Stokes equations, the standard analytical approach, and I believe there is another way, a simpler geometrical approach. As an aside, my geometrical approach is also "analytical" in the broad definition in that it provides an "analysis" leading to a quantitative solution.

My "misplaced antagonism" about van der Veen's "equation formatting" was based on a version of his 2016 paper provided by my colleague James Fastook that had those formatting defects. The defects were missing in the version that actually appeared in The Cryosphere, as I subsequently acknowledged when I saw that version. So let's put

this "objection" to rest.

What remains is important. Van der Veen's Figure 3 misrepresents my geometrical force balance. There is no other word to describe it. I can only conclude he doesn't understand the geometrical force balance (which is my fault; I didn't present it clearly enough), or he was just careless by not making his lines AF and BE parallel. We all make careless mistakes.

Van der Veen made a careless mistake in his 31 January 2017 response to my reply: his Equation (1) contains a longitudinal force gradient along flow direction x, not a longitudinal stress gradient, yet his plot of that equation in his Figure 2 labels it "gradients in longitudinal stress". So which is it? And how does he sort out a "gradient in longitudinal stress" from his Equation (1) anyway? Differentiating his force gradient by parts gives a longitudinal stress gradient times ice thickness plus a longitudinal stress times an ice thickness gradient.

His Equation (2) has another careless mistake. It presents the Navier-Stokes equation written for direction x of ice flow, yet the last term does not contain x, only the yz stress and the z direction. It should be the zx stress in the z direction.

Once again, Van der Veen insists "Flow of glaciers is driven by gradients in the gravitational lithostatic stress that can only be estimated over a certain horizontal distance." His conclusion is based on his Equation (1) which presents stresses after integrating the Navier-Stokes stress-gradient equations. Force gradients in his Equation (1) are stresses. Weertman (1957) obtained the force balance at the calving front of a flat ice shelf by integrating these equations, but Robin (1958) obtained the very same solution geometrically with no gradients in gravitational stresses over no horizontal distances. His force balance is obtained at x = 0. Van der Veen will have to explain, very carefully so everyone can understand, why it is impossible to get the same solution using the two different approaches, with and without stress gradients.

Here it is worth emphasizing that Van der Veen accepts my geometric force balance for

linear shelf flow and linear sheet flow. He only objects to my solution for linear stream flow, which readily connects sheet flow to shelf flow by progressively uncoupling ice from the bed by progressively drowning the bed to produce a floating fraction of overlying ice, my variable phi. My phi produces the typical concave profile of ice streams in a very direct way, where he struggles with convoluted "bridging stresses" over cavities without specifying a water pressure (if any) in his cavities. I do specify the water pressure in my floating fraction.

Van der Veen also needs to explain, using his "force budget" approach to solving the Navier-Stokes equations, how it produces water pressure in his subglacial cavities that pushes water far above sea level in West Antarctic ice streams, as Kamb and Engelhardt demonstrated by measuring water heights in many boreholes. My geometrical approach produces this water height directly from the floating fraction of ice along an ice stream calculated from known ice thicknesses and surface slopes. You can't beat that.

I agree with Van der Veen on one point. "It is always a good idea not to put words in someone else's mouth." My interpretation of his Figure 4a, as it is drawn, does not allow horizontal spreading, as each horizontal arrow at each depth cancels the opposite arrow. That's all I said. Does he deny that? We both know how to draw a geometrical figure that allows horizontal spreading of a flat ice shelf (or ice divide), see Weertman (1957) and Robin (1958), and our books, mine in 1998 and 2012 and his in 1999 and 2013.

---

## Author Comment (AC3) · 18 Mar 2017

Anonymous Referee #2 deserves a reply. He states, "Having read Hughes' criticisms, I still cannot find fault with Van der Veen's approach, his reasoning, or his conclusions." I also find no fault–when Van der Veen is discussing his analytic approach to the force balance based on integrating the Navier-Stokes equations. I find fault with his characterization of my geometric approach, which is probably my own fault for not making my approach clear enough for him. Anonymous Referee #2 may also be confused. So I'll try again.

My geometrical approach to the force balance for linear ice streams began in 1988 when I introduced my concept of the pulling power of ice streams quantified by my

floating fraction "phi" of the bed under an ice stream that was covered by water deep enough to drown bedrock projections into the ice (and, by extension, to supersaturate subglacial till so it also offered no resistance to glacier sliding). After extensive review, that work was published by the Journal of Glaciology in 1992: On the pulling power of ice streams, 38(128), 125-151.

My pulling power paper introduced my vertical force balance using floating fraction phi to determined the height up a borehole basal water pressure would drive basal water. That made it possible to map the floating fraction of ice at the bed by mapping the height of basal water in boreholes, a task undertaken by Barclay Kamb and Hermann Engelhardt several years later for West Antarctic ice streams, notably Kamb Ice Stream. Nothing more was needed to produce those maps. Integrating the Navier-Stokes equations cannot deliver that result.

At the same time, in 1988, I introduced my geometrical approach to the horizontal force balance in the direction of ice flow. This was a tougher nut to crack and several attempts to get it right were presented in papers of mine spanning the next 24 years when Nova published my book, Holistic Ice Sheet Modeling: A First-Order Approach, in 2012. That book presents two solutions, a robust approximate solution based only on the longitudinal force balance, and a more accurate solution based on both the force balance and the mass balance (both solutions good for flowlines and for flowbands of constant width).

My approximate solution is Van der Veen's Equation (12) in his 2016 paper. It is plotted in his Figure 1 (bottom) for Byrd Glacier. It has the considerable advantage of giving a good approximation to the floating fraction over the whole ice drainage basin of an ice stream consisting of tributaries that converge on the ice stream, without actually having to know the paths of tributaries. Specifying phi at any location specifies ice elevation above the bed at that location. This is a boon for reconstructing former ice sheets from glacial geology that doesn't show changing flow directions through time, see J.L. Fastook and T.J. Hughes, New perspectives on paleoglaciology, Quaternary

Science Reviews, 80, 169-194, 2013.

The more accurate solution overcomes the main defect of the approximate solution, namely, basal water pressure cannot push basal water higher than its height above the bed at sea level (or lake level) when the ice stream becomes totally afloat. Hence, floating fraction phi is tied to the ice thickness where an ice stream meets an ice shelf or becomes wholly afloat as an ice tongue. In the more accurate solution, basal water pressure pushes basal water up boreholes to the heights observed by Kamb and Engelhardt for West Antarctic ice streams, see B. Kamb, Basal zone of the West Antarctic ice streams and its role in lubrication of their rapid motion. In Alley, R.B., and Bindschadler, R.A. (Eds.), The West Antarctic Ice Sheet: Behavior and Environment, American Geophysical Union, Antarctic Research Series, Washington, D.C., pp. 157-200, 2001.

Fortunately, the robust solution for phi is sufficient for most ice streams draining past and present ice sheets. The more accurate solution is needed for West Antarctic ice streams because they are unusually long and low, owing the the advanced stage of gravitational collapse of the West Antarctic Ice Sheet.

Ice-stream thickness at an ice-shelf grounding line is linked to the ability of a laterally confined and locally pinned ice shelf to buttress ice streams that supply it with ice. The more buttressing, the thicker the ice, and therefore the higher the water column at the grounding line. This allows phi at any location drained by the ice stream to push water higher in boreholes. The product of these two values of phi, one at the grounding line and one in the ice drainage basin (combining the more robust and more accurate solutions) produces a basal buoyancy factor that allows ice streams to be assigned a position in their life cycle, from inception to growth to mature to declining to terminal. This has been done for all the major Antarctic ice streams at the present time, see T. Hughes, A simple holistic hypothesis for the self-destruction of ice sheets, Quaternary Science Reviews, 30(15-16), 1829-1845, 2011. This kind of insight and synthesis is not possible using only the Navier-Stokes equations in the force balance.

I hope, as a result of this exchange, I will have an opportunity to improve my reply to the 2016 paper by Van der Veen, especially by showing how both the vertical and longitudinal force balances are needed to quantify floating fraction phi so that measuring the height of water in boreholes allows the regions of ice-bed uncoupling to be mapped accurately under an ice stream and along tributaries supplying it. This is a major challenge that needs a solution in glaciology. A first step comes from the geometrical force balance.

It is always a mistake to embrace the "settled science" mantra. Integrating the Navier-Stokes equations is not the end of the story in understanding the force balance in glaciology, or in modeling ice sheets using the force, mass, and energy balances to obtain three dimensional numerical solutions for changing ice dynamics over time. That puts science in a box. Advances in science come from thinking outside the box.

---

## Author Comment (AC4) · 22 Mar 2017

May I add one more comment regarding Van der Veen (2016) and the subsequent exchanges, including comments by Anonymous Referee #2? Van der Veen (2016) states his Equations (13) through (15) containing my floating fraction "phi" are my equations, but they are not. My equations for these stresses are given in Table 1 of my 12 January 2017 reply to Van der Veen (2016). Both signs and terms in his equations are different from those in my equations. Also, his Equation (9) for the force balance has signs different from the signs in my geometrical force balance. The easiest demonstration of this is to substitute his Equations (13) through (15) into his Equation (9) and you will not get 0 = 0. When my correct equations are substituted into my correct geometrical force

balance equation, 0 = 0 is obtained. So it is no wonder his Figure 2, which is his plot of these equations, makes no sense. Had I made these substitutions earlier myself, much of these exchanges could have been avoided, but I appreciate the exchanges as a way to more clearly present the geometrical force balance.

---

## Editor Comment (EC1) · F. Pattyn (Editor) · 31 Mar 2017

This reply is a re-iteration in the defense of the analytical force balance that does not add anything new to the discussion launched by C. van der Veen. Both authors (Hughes, 2016 and van der Veen, 2016) make their point in defending and vividly demonstrating the usefulness of their approaches. Having the two papers published in TC definitely makes comparison between both approaches possible for TC readers.

In this reply, Hughes addresses a number of errors that are clearly not present in van der Veen (2016) and probably relate to a version of the manuscript that is not published. As correctly stated by one of the referees, there is nothing wrong with the analysis of van der Veen. Moreover, the reply contains a lot of anecdotal reference that may be of

interest for some (or of potential interest from a historical perspective), but are out of place in a reply on a published manuscript. Furthermore, the tone of the reply as well as its format is not what is to be expected for a scientific reply within a scientific journal. Referee 2 summarizes this neatly.

Given that the discussion around this reply is at certain stages too personal and with limited interest for TC readers , I suggest that the reply by Hughes and the associated discussion remains part of the legacy of TC discussions (as it is a discussion) and does not get published as a final TC paper in The Cryosphere.

---

## Author Response (AR1)

**Reply to "Basal buoyancy and fast-moving glaciers: in defense of analytic force balance" by C. J. van der Veen (2016)**

Terence J. Hughes, Professor Emeritus of Earth Sciences and Climate Change
University of Maine

North Sixth Street, Fort Pierre, South Dakota 57532, U.S.A.

**Abstract.** Two approaches to ice-sheet modeling are available. Analytical modeling is the traditional approach (Van der Veen, 2016). It solves the force (momentum), mass, and energy balances to obtain three-dimensional solutions over time, beginning with the Navier-Stokes equations for the force balance. Geometrical modeling employs simple geometry to solve the force and mass balance in one dimension along ice flow (Hughes, 2012a). It is useful primarily to provide the first-order physical basis of ice-sheet modeling for students with little background in mathematics. The geometric approach uses changes in ice-bed coupling along flow to calculate changes in ice elevation and thickness, using floating fraction $\phi$ along a flowline or flowband, where $\phi = 0$ for sheet flow, $0 < \phi < 1$ for stream flow, and $\phi = 1$ for shelf flow. An attempt is made to reconcile the two approaches.

**Introduction**

Cornelis "Kees" Van der Veen's comparison of geometric and analytic approaches to the force balance in glaciology in *The Cryosphere* (Van der Veen, 2016) is most welcome because he takes seriously my geometrical approach to the longitudinal force balance, citing many of my paper from when I first introduced the concept (Hughes, 1992) to the latest application (Hughes et al., 2016). To begin, the analytic force balance is not challenged by me. The geometric force balance is useful only for one-dimensional flow along ice-sheet flowlines or flowbands of constant width. For two-dimensional flow in the map plane, width become a variable and geometrical areas become geometrical volumes; substantially increasing geometrical complexity with little advance in physical insight. The analytic force balance is typically obtained by solving the Navier-Stokes equations, which can be done in three dimensions and, when including the mass and energy balances, becomes time-dependent. The geometrical approach is useful for understanding the force balance by comparing the areas of right triangles and rectangles (or parallelograms).

**Addressing Van der Veen (2016)**

My interest in the force balance for ice sheets spans four decades, beginning when I used glacial geology to reconstruct former ice sheets from the bottom up based on the strength of ice-bed coupling deduced from glacial geology, an approach that also produced the concave surface of ice streams for the first time (Denton and Hughes, 1981, Chapters 5 and 6). I developed the geometric approach after observing the huge arcing transverse crevasses at the head of Byrd Glacier, and realized it was actually pulling ice out of the East Antarctic Ice Sheet (Hughes, 1992). Since then it has been a work in progress. Van der Veen (2016) cites earlier stages of that work (Hughes, 2003, 2008). I would prefer that he use my current treatment in Hughes (2012a) and Hughes et al. (2016).

Referring to Hughes (2008), Van der Veen (2016) states on his page 1332 that I believe
lateral drag vanishes at the center of an ice stream. Lateral shear stress $\sigma_{xy}$ vanishes, but
the lateral shear force does not. On one side, stress $\sigma_{xy}$ acts on side area $A_y$ and on the
other side stress $-\sigma_{xy}$ acts on side area $-A_y$, with $A_y$ and $-A_y$ being vectors in opposite $y$
directions, so the shear force is always positive and opposes longitudinal gravitational
forcing.

Van der Veen (2016) states his Eq. (9) is my Eq. (36) in Hughes (2003). It is not, his
signs are different from mine and his $\sigma_F$ is not the same as my $\sigma_T$. In the geometric force
balance, the driving force is the area of a triangle and all the resisting forces are areas of
triangles and a rectangle (or parallelogram) that fit into the triangle so the driving and
resisting forces are identical. All signs are positive in my Eq. (36). His $\sigma_F$ is my flotation
stress, which doesn't appear in my 2003 paper. It appears in my Nova book, *Holistic Ice*
*Sheet Modeling* (Hughes, 2012a) and in Hughes et al. (2016) in *The Cryosphere*. Van der
Veen (page 1333) states my $\sigma_F$ is his $\tilde{R}_{xx}$. It is not. His force budget approach has no way
for calculating my flotation stress $\sigma_F$ because his approach has no place for my floating
fraction $\phi$ of ice under an ice stream (which he calls a "basal buoyancy factor" that
obscures its physical meaning), see my Fig. 1.

Van der Veen (2016) states his Eqs. (13), (14), and (15) are my equations in my 2008,
2012a, and 2016 publications. They are not. His signs are different from mine and even
some of his terms are different from mine. The proof is found by substituting his Eqs. (13)
through (15) into his Eq. (9), which does not deliver 0 = 0 for the force balance. My
equations, reproduced as my Table 1 from Table 12.1 in Hughes (2012a), do give 0 = 0. In
my geometric force balance, resisting forces are represented by triangles and a rectangle
(or parallelogram) that exactly fit inside a big right triangle that represents my driving
force, so the area of my big triangle is the same as summed component areas from resisting
forces within it. Therefore 0 = 0 *must* be obtained, see my Fig. 2.

Van der Veen (2016) plots his Eqs. (9) through (15) in his Fig. 2, so they cannot
represent my force balance because they are not my equations. Also the plot of his
"Gradients in longitudinal stress" should be gradients in longitudinal force, which is a
stress, so he can compare stresses with stresses, not with stress gradients of stresses. If his
Fig. 2 truly plots a longitudinal stress gradient, it compares apples with oranges. Also in his
Fig. 2, his longitudinal stress (or force) gradient acts in the same direction as his
gravitational driving force. That is impossible in my geometric force balance, see my Fig. 2.

Referring to my Figure 3 (left), Figure 3 in Van der Veen (2016), line AF should be
parallel to line BE because they both show ice pressure increasing linearly with depth. Line
CE shows how water pressure increases linearly with depth, as is obvious at the calving
front. In my geometrical force balance, the longitudinal gravitational driving force is area
ADF of the big triangle. Fitted inside ADF are a resisting flotation force given by area BDE
for floating ice fraction $\phi$ and a resisting drag force given by area ABEF for the grounded
ice fraction $1-\phi$ in my Fig. 1. Inside BDE is area CDE for the resisting force from water pressure and area BCE for the resisting force from the tensile strength of ice. Inside area
ABEF is the triangle above B for basal drag and the parallelogram below B for side drag.
Resistance from basal drag is the area of the triangle above B. Resistance from side drag is
the area of the parallelogram below B if lines BE and AF are made parallel. If BE is made
part of AF a rectangle would replace the parallelogram but the area would be unchanged,
see my Fig. 2. That's all there is to it. The only remaining task is to replace forces with
products of stresses and lengths (for areas having unit or constant widths along $x$) upon
which the stresses act along a flowline (no width) or a flowband (constant width). My
solution for the force balance is exact because forcing area ADF equals resisting areas
ABEF, BCE, and CDE inside ADF. All gravitational and resisting forces in the longitudinal
direction of ice flow are thereby included, with ABEF representing the force from both
basal and side drag.

Van der Veen (2016) correctly states his Eq. (16) represents my longitudinal
gravitational driving force, but then he states it "does not represent the gravitational
driving force" (page 1335). It does. In my direction $-x$ of ice flow, the gravitational force (a
horizontal vector) is the average ice pressure (a scalar) times the transverse cross-
sectional area against which it acts (as a horizontal vector in my $-x$ direction), which for
an ice stream of constant width is ice width times ice height above the bed, a height that
varies along $x$, as does average ice pressure, so the gravitational driving force varies along
$x$. The correct representation of my longitudinal geometric force balance is my Fig. 2 where
his area ABEF is my area 1+2 for basal and side drag at $x$.

Van der Veen (2016) states on his page 1335 that a longitudinal force balance along $x$
must be made over incremental distance $\Delta x$ that shrinks to zero. My longitudinal force
balance along $x$ *does* in my Fig. 2 (bottom), see Hughes (2012a, Appendix G) and Hughes et
al. (2016, page 10). I subtract longitudinal force areas over distance $\Delta x$ to get my
longitudinal force balance Eq. (22) in Hughes et al. (2016). However, Van der Veen (2016)
is incorrect in stating a longitudinal force balance *always* must be made over length $\Delta x$. At
the calving front of an ice shelf the balance is obtained right at the calving front where
$\Delta x = 0$, as Robin (1958) proved 59 years ago *geometrically*.

Van der Veen (2016) discusses areas ADF and APD in terms of "lithostatic stresses"
increasing with depth in his Fig. 4(a), shown in my Fig. 3 (right). The areas are forces. As he
shows by his horizontal arrows in his Fig. 4(a), area ADF is my horizontal gravitational
driving force and area APD is the sum of my horizontal resisting forces opposing the
driving force in my geometrical force balance shown in my Fig. 2 (center) with an ice
surface slope at $x$. His area APD can be subdivided into my smaller areas of triangles and a
rectangle in my Fig. 2 (center) to obtain areas that resist gravitational forcing from his area
ADF. There is no surface slope in his Fig. 4(a), a condition that applies to an unconfined
linear ice shelf having constant thickness (Weertman, 1957; Robin, 1958), in which case
only my areas 3 and 4 in my Fig. 2 (bottom) add to give his area APD since there are no
basal and side drag forces represented by my areas 1 and 2. Raymond (1982) analyzed
deformation near interior ice divides where the surface slope is also zero.

Van der Veen (2016) correctly shows the geometrical force balance in my Fig. 2
(bottom) for a sloping ice surface above a horizontal bed in his Fig. 4(b), shown in my Fig. 3

(right). From these figures we can both obtain the geometric longitudinal force balance
over incremental length $\Delta x$ in analytic form when $\Delta x \to 0$. In my Fig. 2 (bottom), my big
triangles at $x$ and $x + \Delta x$ are gravitational driving forces that are respectively subdivided
into areas 1, 2, 3, 4 and areas 5, 6, 7, 8 that resist gravitational motion along $x$.

**My Geometrical Force Balance**

I developed the geometrical force balance to teach the fundamentals of glaciology to
students with an inadequate background in mathematics, usually students studying to be
glacial geologists (Hughes, 2012a). My geometrical approach was designed to make
maximum use of glacial geology in reconstructing former ice sheets from the bottom up
(Hughes, 1998, Chapters 9 and 10; Fastook and Hughes, 2013) and in demonstrating how
basal thermal conditions produce glacial geology under the Antarctic Ice Sheet today
(Hughes, 1998, Chapter 3, Wilch and Hughes, 2000; Siegert, 2000). Previously I had spent
more time teaching calculus than glaciology because the Navier-Stokes equations had to be
integrated in the force balance.

The major variable in my geometrical force balance is the floating fraction $\phi$ of ice,
where $\phi = 0$ for sheet flow, $0 < \phi < 1$ for stream flow, and $\phi = 1$ for shelf flow. Here we are
primarily interested in stream flow as shown in my Fig. 1 for possible $\phi$ distributions at
the bed and my Fig. 2 for the longitudinal force balance. From Newton's second law of
motion in a vertical force balance, gravitational force $F_G$ at the base must be the same for
floating area $w_F \Delta x$ and total area $w_I \Delta x$ such that $F_G = (\rho_I h_I w_F \Delta x)g = (\rho_I h_F w_I \Delta x)g$ for ice
density $\rho_I$ and gravity acceleration $g$ to obtain basal pressures $P_F = \rho_I g h_F$ and $P_I = \rho_I g h_I$
that support ice of respective floating and total heights $h_F$ and $h_I$. This vertical force
balance is satisfied if $h_F$ goes from 0 to $h_I$ as $w_F$ goes from 0 to $w_I$. The basal water
pressure is $P_W = \rho_W g h_W = P_F = \rho_I g h_F$ for water density $\rho_W$ and water height $h_W$ needed to
float ice height $h_F$. The floating fraction of ice at $x$ is therefore:

$$\phi = w_F / w_I = h_F / h_I = P_F / P_I = P_W / P_I.$$

Pulling force $\sigma_T h_I$ resists the gravitational driving force given by area 4 in Figure 2
(bottom), which is area 3+4 minus area 3. Area 3+4 is one-half flotation height $h_F = h_I \phi$
times basal floating length $P_F = P_I \phi$, so area 3+4 is $\bar{P}_I h_I \phi^2$. Area 3 is one-half height
$h_W = (\rho_I / \rho_W) h_F = (\rho_I / \rho_W) h_I \phi$ times the same basal floating length $P_F = P_I \phi$. Then the
tensile pulling stress is $\sigma_T = \bar{P}(1 - \rho_I / \rho_W)\phi^2$. It is that simple. At the calving front where
$\phi = 1$ this is the solution obtained by Weertman (1957) and Robin (1958). Table 1 lists all
stresses resisting gravitational forcing at $x$.

At distance $x$ from the ice-shelf grounding line in my Fig. 2, gravitational driving force
$F_G = \bar{P}_I h_I$ is resisted by the sum of upstream tensile pulling force $F_T = \sigma_T h_I$ and
downstream compressive pushing force $F_C = \sigma_C h_I$ so $\sigma_T = \bar{P}_I - \sigma_C$. Tensile force $\sigma_T h_I$
balances the part of the driving force equal to area 4, and resisting force $\sigma_C h_I$ balances the part of the driving force equal to areas 1+2+3 in Figure 2 (center and bottom), and includes
all downstream resistance due to averaged basal and side shear stresses $\bar{\bar{\tau}}_O$ and $\bar{\bar{\tau}}_S$
respectively linked to areas 1 and 2, plus local water buttressing stress $\sigma_W$ linked to area 3,
all of which resist gravitational forcing equivalent to these areas.

[revised manuscript text omitted]

---

## Author Response (AR2)

Dr. Frank Pattyn, Editor, The Cryosphere

Dear Frank:

The version of tc-2017-6 submitted here addresses concerns you expressed to me on 18 May 2017. Following your instructions, I have removed all derivations and related material that do not directly address Van der Veen (2016). The abstract and text are both shortened and more focused. In particular I paid attention to your point that his approach and mine are similar but not identical, so each has to be assessed on its own terms. An attempt to somehow blend the two approaches leads to confusion. I agree. Thank you for your own insights.

Much appreciated,

Terry Hughes

Fort Pierre, South Dakota, USA, 20 June 2017